# ‘A Healthy CIT’: An Investigation into Student Health Metrics, Lifestyle Behaviours and the Predictors of Positive Mental Health in an Irish Higher Education Setting

**DOI:** 10.3390/ijerph16224318

**Published:** 2019-11-06

**Authors:** Andrea Bickerdike, Joan Dinneen, Cian O’Neill

**Affiliations:** Department of Sport, Leisure & Childhood Studies, Cork Institute of Technology, Cork T12 P928, Ireland; joan.dinneen@cit.ie (J.D.); cian.oneill@cit.ie (C.O.)

**Keywords:** university students, health, gender, health promotion, health behaviours, lifestyle, healthy universities, BMI, mental health

## Abstract

Higher Education Institutions (HEIs) are potent health promotion settings, uniquely positioned to aid societal efforts to combat non-communicable diseases (NCDs). International evidence suggests that health metrics and lifestyle behaviours of higher education students are sub-optimal, yet a dearth of contemporary Irish data exists. This study aimed to examine sex differences in student lifestyle behaviours and identify significant predictors of positive mental health in an Irish HEI setting. An online questionnaire instrument distributed to all registered students (*n* = 11,261) gathered data regarding a multitude of health and lifestyle domains. Many items were adapted from previous Irish research. Further validated scales included the Alcohol Use Disorders Identification Test (AUDIT), Mental-Health Index 5 (MHI-5) and the Energy and Vitality Index (EVI). Self-reported height/body mass were also recorded. In total, 2267 responses were analysed (51.7% female, 48.3% male). Both sexes demonstrated poor sleeping patterns, hazardous drinking and sub-optimal fruit and vegetable intake. The calculated prevalence of overweight/obesity was 38.2%. Both sexes underestimated obesity. Males underestimated and females overestimated overweight. Males displayed riskier behavioural patterns with regard to illicit substances, drinking, and sexual partners. Females reported greater psychological distress. Multivariate linear regression identified 8 variables as predictors of positive mental health, accounting for 37% of the variance in EVI scores. In conclusion, HEI students would benefit from sex-specific multi-level health promotion initiatives to remove macro-level barriers to healthier lifestyles.

## 1. Introduction

Unhealthy lifestyles increase the risk of non-communicable diseases (NCDs) such as cardiovascular disease, respiratory disorders, diabetes and cancers [1]. In 2016, NCDs accounted for 71% of worldwide deaths and 91% of population deaths in Ireland [2]. Due to the deleterious physiological and economic effects of NCDs, enabling healthy behaviours throughout the lifespan is a fundamental priority of the Irish Government’s health promotion strategy [3]. 

In this regard, higher education may constitute a cost-effective setting for the implementation of health education initiatives [4]. It is a uniquely transitional life stage where a multitude of lifestyle behaviours and social experiences are interwoven within the culture and built environment of a single Higher Education Institution (HEI) setting [5]. However, financial scarcity and academic stressors [6] experienced within a newly autonomous environment can result in students engaging in risk-taking behaviours [7] that are associated with the quintessential ‘college lifestyle’. 

In addition, the ‘Freshman 15’ (an alleged weight gain of 15 pounds or 6.8 kg during a student’s first year of higher education) has remained a widely-cited phenomenon, despite a lack of objective verification [8]. A recent meta-analysis incorporating data from 5549 higher education students reported that the overall mean gain in body mass was 3 pounds (1.36 kg), 60.9% of students gained at least some weight and 9.3% gained the postulated ‘freshman 15’ [9]. In a general context, the volume of studies reporting increases in students’ body mass during the first year of higher education [10,11,12,13,14] constitutes a worrying physiological trajectory. 

Similarly, detrimental trends in body composition have also been observed over the course of a four-year academic programme in a USA university, with significant increases in students’ Body Mass Index (BMI), body fat percentage and total fat mass values [15]. Using BMI as an indicator of adiposity, a study investigating overweight/obesity among HEI students across seven European countries reported prevalence ranges of between 4.6% (Bulgaria) to 18% (Denmark) among females and 12% (Lithuania) to 27.3% (Germany) among males [16]. This study also highlighted generalised misperceptions of anthropometric status; male students classified themselves as too thin whereas females perceived themselves as excessively fat [16]. 

HEI students also appear to exhibit sub-optimal dietary behaviours as evidenced by inadequate intake of fruit and vegetables [7,17,18,19,20,21] and habitual consumption of fast food during a typical week [22,23,24]. In particular, students who move away from home appear to consume fewer home-cooked meals [20]. Reported breakfast consumption patterns have been variable between studies. A German study reported that just over one quarter did not regularly (defined as 4–5 per week) eat breakfast [17] whereas research conducted in the USA reported that 38.7% of students consumed breakfast less than five times per week [25]. The transition from school to university may also be a time where students become less physically active, particularly those who move residence [26]. It appears that eating and physical activity behaviours are influenced by both personal motivations and self-regulation, further complexified by the unique barriers interwoven within the campus environment and social constructs of the college lifestyle [27].

Hazardous drinking habits have been widely documented among HEI students in Ireland [28], the United Kingdom [29,30], Belgium [31], Australia [32], New Zealand [33,34], North America [35] and South Africa [36]. In addition, although the exact prevalence of drug use among young people is difficult to establish due to a scarcity of objective studies [37], illicit substance misuse has been reported among cohorts of higher education students, both in terms of prescription medications [38] as well as recreational substances [24,37]. In terms of sexual health, a casual ‘hook-up’ culture has been reported, with students displaying a lack of concern regarding their vulnerability to sexually transmitted infections [39]. 

Psychologically, inherent stressors such as extensive academic workloads, scarce finances and uncertainty regarding the future are ingrained within the very nature of the higher education experience [40]. Previous research has reported the prevalence of psychosomatic symptoms such as fatigue (59.7%), headaches (57.8%) and back pain (42.7%) among a cohort of higher education students [6]. Psychological wellbeing may be further attenuated by poor sleep quality [41] which can be a normative aspect of university life [42]. 

From a wider socio-cultural perspective, technological advances and the emergence of mobile technologies over the past number of years may have influenced contemporary student lifestyle behaviours and parameters of wellbeing. For example, smartphone use may be associated with poorer sleep quality in a relationship mediated by depression and anxiety [43], and internet addiction has emerged as a health concern among first year university students in Turkey [44]. Social media has become ingrained within the lives of many young adults [45] and exposes users to alcohol-related content and advertising [46]. 

Traditionally, studies pertaining to student lifestyles have emphasised a particular health domain or behaviour [21]. However, it has been previously argued that health-related behaviours should not be studied in isolation but as collections that can cluster in favourable or unfavourable patterns [47]. In addition, it is possible that institution-/university-specific clustering of either ‘health-promoting’ or ‘health-damaging’ indicators exist, hence the need to establish setting-specific profiles from which to guide campus health promotion programmes [6].

Demographically, sex differences in the lifestyle behaviours of higher education students have been elicited. Males have been shown to be more physically active [7,21,47,48,49], however greater levels of illicit drug use have also been reported [24]. Females have consistently reported greater levels of psychological stress [6,24,47], but have also exhibited a tendency to consume more portions of daily fruit and vegetables [7,47]. Interestingly, traditionally accepted sex differences in drinking patterns may have changed with recent Irish research reporting a greater prevalence of hazardous drinking among females [28].

It is evident that in any effort to describe the contemporary lifestyle patterns of higher education students, sex must be considered as a demographic covariate. Secondly, lifestyles must also be considered within the cultural context of the HEI setting. Finally, ongoing research is required to document student lifestyle patterns over time. In Ireland, a designated longitudinal programme of research was recommended following the worrying lifestyle trends identified by a nationally representative study of undergraduate students [24]. However, such a multi-setting programme of research was never implemented and a relative dearth of contemporary data exists. 

The purpose of the current study was to explore a series of student health parameters and lifestyle behaviours within a medium-sized HEI in southern Ireland. Specific objectives focused on (1) the examination of sex-related differences across a number of health and lifestyle domains and (2) the identification of significant lifestyle predictors of positive mental health. Findings will be utilised to identify pertinent action areas for a new campus health promotion initiative (*A Healthy CIT*) and will be of interest to policy makers in similar institutions elsewhere. 

## 2. Materials and Methods

### 2.1. Study Design

*A Healthy CIT* is a campus health promotion initiative at Cork Institute of Technology (CIT) that aims to maximise the health and wellbeing of all students and staff. Launched in 2016, the initiative endeavours to integrate a consideration for health and wellbeing into all aspects of the Institute’s operations, in accordance with the recommendations of the Okanagan Charter for Health Promoting Universities and Colleges [50]. *A Healthy CIT* encompasses a designated research arm within CIT’s Department of Sport, Leisure and Childhood Studies to strategically guide its future development and evaluation. Development and pilot activities are described in further detail elsewhere [51]. 

Data included in this study were collected as part of baseline research activities; a cross-sectional web-based health survey conducted during Semester Two (Spring) of a standard academic year. All registered students of the Institute, aged 18 and over, were eligible to participate. The study was conducted in accordance with the Declaration of Helsinki, and the protocol and questionnaire instrument were approved by the Research Ethics Committee of the host institution prior to data collection. 

### 2.2. Questionnaire Instrument

A questionnaire instrument was developed consisting of 92 main items pertaining to a series of health and lifestyle domains (demographics, general health, food habits and nutrition, physical activity, alcohol, tobacco and drug use, sexual health, sleep patterns, mental health, and social media use). Many items were adapted from previous Irish research [24,52,53,54] to facilitate retrospective comparisons. The instrument was hosted on an online platform (‘Lime Survey’). Skip-logic exposed participants only to relevant items based on their previous responses. Variables included in the current analyses were: 

*Demographics:* sex, age, nationality, year of study, National Framework of Qualifications (NFQ) level (‘undergraduate/taught postgraduate/postgraduate by research’), mode of study (‘full time student/ part time or evening student’) and area of study (later recoded by Faculty). These items were in line with previous Irish research [24,52,53]. 

*Academic achievement* during the preceding semester was self-reported on a categorical item reflecting the Institute’s grade bands (less than 40%/40–59%/60–69%/70% or above/don’t know/I would rather not provide this information). 

*Self-reported general health* was measured using a single item adapted from previous Irish research [24,54] (‘In general, would you say your health is’) with answers on a 5-point Likert Scale: (1) ‘very poor’, (2), ‘poor’ (3) ‘neither poor nor good’ (4) ‘good’ (5) ‘very good’. ‘Poor’ and ‘very poor’ categories were later combined for analysis. 

*Body Mass Index [BMI]* was calculated from self-reported height (‘What is your estimated height?) and body mass (‘What is your estimated current weight?’) using the standard formula: $\frac{Weight\;\left(kg\right)}{{Height}^{2}\;(m^{2})}$. BMI weight categories were defined according to World Health Organisation criteria; <18.5 kg/m^2^ = underweight, 18.5–24.9 kg/m^2^ = normal weight, 25.0–29.9 kg/m^2^ = overweight and ≥ 30.0 kg/m^2^ = obese [55].

*Self-perceived weight category* was measured with a single categorical item (‘Which general category do you feel best describes you?’). Participants selected either ‘underweight’, ‘normal weight’, ‘overweight’ or ‘obese’. 

*Daily fruit and vegetable servings* were self-reported (‘How many servings of fruit and/or vegetables do you eat on average every day?’). A reference description of a serving was provided as ‘3 dessert spoons of vegetables or 1 piece of fruit’ [52]. 

*Habitual breakfast consumption* was measured with two items pertaining to consumption during the week (‘How many days during the week do you normally eat breakfast?’) and at weekends (‘How many days at the weekend do you normally eat breakfast?’). Participants were required to self-report numerical values from 0–5 (weekdays) or 0–2 (weekends).

*Barriers to physical activity* were elicited with a single categorical item sourced from previous Irish research [54] (‘What would you say is the main reason why you are not (more) physically active at this time?’). Participants were provided with seven potential barriers, as well as the option to specify an additional ‘other’ barrier. A ninth barrier (‘I feel that I may be taking too much exercise or overtraining’) was added to this item specifically for the current study. 

*Physical activities (type) during the 7 days prior to survey participation* were elicited using three newly-devised items each pertaining to (i) light, (ii) moderate and (iii) vigorous intensity activities respectively. These items were formulated based on the structure of the International Physical Activity Questionnaire [56] and previous Irish research [24]. For each item, participants were provided with a reference description of the relative intensity. A list of activities and popular sports was then provided, based on sample activities listed in the Irish physical activity guidelines [57] as well as the 2015 report of the Irish Sports’ Monitor [58]. Participants were instructed to select all sporting, routine or occupational activities (if any) that they had participated in.

*Physical Activity (volume) during the 7 days prior to survey participation* was quantified using a newly-devised item that required participants to enter (i) the frequency and (ii) duration (in min) of each session of light, moderate and vigorous physical activity undertaken (if applicable). This enabled the calculation of total volume of each respective category in min per week (frequency x duration). The primary variable of interest was whether students were sufficiently physically active according to Irish National Physical Activity Guidelines for adults [57]. A participant was deemed to have met the guidelines if their combined volume of vigorous and moderate activity reached a threshold of 150 min of moderate intensity activity. For the purposes of these calculations, 1 minute of vigorous intensity activity was considered to be equivalent to 2 min of moderate intensity activity, as outlined in the aforementioned Irish National Physical Activity Guidelines document [57].

*Daily sitting time* was measured with a newly devised ordinal item (‘During an average day at college for you, how many hours do you usually spend sitting down? For example, sitting at a desk, driving a car, sitting in a lecture etc.’) with a 6-point scale ranging from ‘less than 1 hour’ to ‘more than 5 h’. 

*Alcohol consumption patterns* were evaluated using the Alcohol Use Disorders Identification Test (AUDIT), a validated 10-item instrument to detect problem drinking [59]. Possible scores range from 0 to 40 with scores between 8–15 warranting advice regarding reduction of hazardous drinking, scores between 16 and 19 suggestive of problem drinking and scores of 20 or more justifying further evaluation for dependence on alcohol [29,59]. The instrument is comprised of a series of subdomains specifically measuring hazardous alcohol use (also referred to as the ‘AUDIT-C’ scale) (items 1–3), dependence symptoms (items 4–6) and harmful alcohol use (items 7–10).

*Tobacco smoking* items were sourced from previous Irish research [24,52] and elicited current smoking status ‘Do you smoke tobacco now?’, with answer options being ‘no’, ‘yes regularly’ or ‘yes occasionally (usually less than 1 per day’). 

*Cannabis/marijuana use* was measured using the following item; ‘On how many occasions, if any, have you used marijuana (grass, pot) or cannabis (hash, hash oil)?’ [52]. Frequency of use was reported on a 7-point Likert scale (never, once or twice, 3–5 times, 6–9 times, 10–19 times, 20–39 times, 40 times or more) with three separate sub-items to distinguish between (i) lifetime use, (ii) use within the 12 months prior to the survey and (iii) use within the 30 days prior to the survey.

*Illicit Substance use* was measured by providing participants with a list of recreational substances (plus their colloquial street names). For each substance, they were requested to report their frequency of use on a 4-point Likert scale (‘never’, ‘yes but not in past 12 months’, ‘once or twice in the past 12 months’, ‘3 or more times in the past 12 months’). This item was adapted directly from previous research [24] with the addition of ‘head-shop products’. The accompanying street names were verified by consultation with local addiction services of the Irish Health Service Executive. The Likert scale was later dichotomised (lifetime use yes/no) for analysis by sex. 

*Sexual health* items were sourced from the Trinity College Survey for Sexual Health, as cited in previous Irish research [24]. Items elicited sexual orientation, sexual activity status, age of first intercourse, use of drugs and/or alcohol prior to previous intercourse, methods of sexually transmitted infection (STI) protection, history of STI diagnosis, morning-after pill use and (self-reported) total number of sexual partners. 

*Self-rated mental health* was measured with a single item (‘How would you rate your own mental health?’) [24] with an identical 5-point response scale as outlined for general health above. 

*Positive and negative mental health* were measured using the Energy and Vitality Index (EVI) and the Mental Health Index-5 (MHI-5) respectively, both of which are subscales of the Short-Form Health Survey (SF-36) [60].

*Psychological stress* was evaluated with an item from previous Irish research that listed 11 stressors of specific relevance to this cohort (issues such as academic workload, finances and relationships plus an option to select ‘other’) [24]. One additional stressor was included (‘social media’) for the current study. Participants were requested to indicate their perception of each stressor on a four-point scale (‘highly stressed’, ‘often stressed’, ‘not often stressed’ or ‘never stressed’). 

*Recent sleep quality* was self-reported on a single item (‘During the past 30 days, how would you rate your sleep quality overall?’) with identical 5-point response scale as outlined for general and mental health ratings above. This item was adapted from the sleep quality component of the Pittsburgh Sleep Quality Index [61].

*Habitual sleep duration* was measured with an item that asked participants to quantify their sleep duration (‘On average, how much sleep do you get?’) both on weekdays and at the weekends [52]. Answers were provided on a 7-point Likert scale ranging from ‘less than 4 h’ to ‘9 h or more’. 

*Social media* items ascertained the platforms used, as well as the average time spent on social media on weekdays and at weekends. These items were adapted from previous Irish research involving a cohort of medical students [62]. Further newly devised items established whether pictures were posted within 30 days of the survey and if so, whether such pictures were alcohol-related (oneself or another consuming or being around alcohol). 

### 2.3. Survey Implementation and Data Collection Procedure

In line with previous research [24,63], the survey was incentivised with entry into a draw to win an iPad. An initial email invitation, containing a direct hyperlink to the questionnaire was distributed to 11,261 registered CIT student email addresses. This email described the purpose and scope of the study, nature of the questionnaire and assured all participants that participation was voluntary and responses were fully anonymised. Participants could also ‘opt-out’ of any further correspondence, if desired, and those who opted out received no further contact. The survey remained accessible for a 23-day period. A total of three reminder emails were sent to all those who had not completed the questionnaire at days 7, 14 and 20, respectively. The survey was reactivated for a 7-day period following cessation of the Institute’s end of semester examinations. 

### 2.4. Statistical Analysis

Data were exported directly to Microsoft Excel (Microsoft Inc., Redmond, WA, USA) and IBM Statistical Package for the Social Sciences [SPSS] Version 25.0 (IBM Inc., Armonk, NY, USA). Data cleaning encompassed a review of all variables to examine the nature of outlier values and missing data. In the current paper, data are presented relative to the number of valid responses received to each applicable item; for the purpose of regression analyses, a ‘pairwise’ method of exclusion was applied. 

Categorical data were analysed by establishing relative frequencies. Where applicable, specified responses initially placed within the ‘other’ category were re-coded to the relevant categorical option. Missing data points due to survey skip logic were excluded from the relative frequencies reported within each health domain but included as valid (coded 0) for three specific variables in the linear regression model (outlined further below). Numeric data were described using means/standard deviations or median/interquartile ranges as appropriate. Sex differences were elicited using Chi Squared Tests for Independence (for categorical variables) and Mann Whitney U Tests as applicable. Alpha level of significance was set at 0.05. 

Physical activity volume was derived based on reported frequency and duration during the 7 days prior to data collection, as outlined above. Self-reported height and body mass values were first mathematically converted (where necessary) from imperial values to metric equivalents and then used to determine BMI. Cohen’s Kappa was calculated to ascertain the relative level of agreement between calculated and perceived BMI category. Validated scales were scored as per relevant instruction manuals. Multivariate linear regression was used to identify significant predictors of positive mental health.

## 3. Results

### 3.1. Response Rate and Student Demographics

In total, 2390 responses (21.2% response rate) were recorded on the LimeSurvey platform (www.limesurvey/org). Blank datasets (*n* = 121) were removed and two spoiled datasets were excluded at a later stage, leaving 2267 cases (20.1% total response rate). Mean completion time recorded by the survey platform was 18.7 min. Due to the substantial length of the questionnaire, there was a progressive decline in responses observed as well as variability in response numbers to each item. A total of 1541 participants proceeded to the final section of the questionnaire (68.0% of initial sample retained, 13.7% overall response rate). Individual item valid response rates ranged from a maximum of 100% (*n* = 2267) to a minimum of 63.9% (*n* = 1449). 

Just over fifty-one percent (51.7%) of those who initially responded were female (*n* = 1173) and 48.3% were male (*n* = 1094). Reported ages ranged from a minimum of 18 to a maximum of 65 years old however the majority (68.9%, *n* = 1560) were aged between 18 and 23. There was no significant association between sex and progressing to the final section of the questionnaire instrument [χ2 (1, *n* = 2267) = 0.64, *p* = 0.43 Phi 0.02]. Table 1 presents each demographic variable by sex.

### 3.2. General Health and Body Mass Index [BMI]

The majority (78.6% *n* = 1667) of participants rated their general health as either ‘good’ or ‘very good’, 15.8% (*n* = 336) rated it as ‘neither poor nor good’ and 5.6% (*n* = 118) as ‘poor/very poor’. Males reported a more favourable perception of their general health than females [χ2 (3, *n* = 2121) = 20.4, *p* = <0.0005, Cramer’s V = 0.01].

Body Mass Index [BMI] calculations (*n* = 1990) classified 11.0% as ‘obese’, 27.2% ‘overweight’, 58.4% ‘normal weight’ and 3.3% ‘underweight’. Absolute BMI values of male students were significantly greater than female values (Md. 24.3 vs. 23.5 kg/m^2^), U = 441280.0, z = −4.18, *p* < 0.0005, r = 0.09.

There was a statistical association between sex and self-perceived BMI category [χ2 (3, *n* = 2069) = 60.2, *p* < 0.0005, Cramer’s V = 0.17] with a greater proportion of females classifying themselves as ‘overweight’ (31.2% vs. 22.5%) or ‘obese’ (4.1% vs.1.9%) in comparison to males. Figure 1 and Figure 2 depict the sex-stratified distribution of both perceived and calculated BMI categories for those who provided all necessary data. The Kappa measure of agreement value between perceived and calculated BMI categories was 0.41, *p* < 0.0005 (females: 0.50, males 0.32).

### 3.3. Food Habits and Nutrition

The median number of servings of fruit and vegetables per day was 3.0 (IQR ± 3.0). Females (Mean Rank 1057.33, *n* = 1046) reported significantly more daily servings than males (Mean Rank 954.75, *n* = 969, U = 455185.5, z = −4.01, *p* < 0.0005, r = 0.09). In relation to breakfast patterns, 62.2% (*n =* 1241) reported a habitual breakfast consumption pattern of five weekday mornings (Mon-Fri) and 73.9% (*n* = 1466) usually consumed breakfast on each of the two weekend mornings (Sat-Sun). There was no significant association between sex and weekday [χ2 (5, *n* = 1996) = 7.2, *p* = 0.20, Cramer’s V = 0.06] or weekend [χ2 (2, *n* = 1984) = 1.9, *p* = 0.39, Cramer’s V = 0.03] breakfast consumption.

### 3.4. Physical Activity and Sedentary Time

A significantly greater proportion of males reached the recommended physical activity guidelines during the 7 days prior to data collection (74.8% vs. 67.8%) [χ2 (1, *n* = 1480) = 8.5, *p* = 0.003, phi = −0.08]. ‘Personal exercise/gym activities’ were the most commonly reported vigorous physical activities undertaken during this period for both males (42.7%, *n* = 371) and females (38.7%, *n* = 354). In relation to moderate intensity activities, males most commonly cited ‘light jogging’ (40.3%, *n* = 350) followed by ‘walking for transport’ (37.6%, *n* = 327). The two most popular activities reported by females were ‘walking for transport’ (53.6%, *n* = 491) and ‘housework’ (*n* = 48.9%, *n* = 448). 

A lack of time and/or exams/college workload was the most commonly reported barrier to physical activity among both males and females (51.1% and 52.2% respectively) as illustrated in Table 2. In relation to sitting time, almost two-thirds reported that they spend at least four hours sitting down during a typical day at college (65.7%, *n* = 1139) and this behaviour was independent of sex.

### 3.5. Alcohol 

The reported age of first alcoholic drink ranged from 9–35 years of age and there were no significant differences between male (Md. = 16.0, *n* = 718) and female (Md. = 16.0, *n* = 773) drinkers in this regard, [U = 274389.0, z = −0.38 *p* = 0.70 r = −0.01].

To evaluate internal consistency, Cronbach’s alpha was calculated for the total AUDIT instrument as well as each sub-domain (Table 3). Males scored significantly higher on the total instrument (Items 1–10: U = 191722.0, *p* < 0.0005), as well as the hazardous drinking subscale (Items 1–3: U = 210397.0, *p* < 0.0005). A breakdown of total and subscale scores by sex is presented in Table 3 below. 

With specific reference to the hazardous drinking sub-domain, 54.1% of females (*n* = 412) and 54.7% of males (*n* = 387) reached previously applied sex-specific threshold scores (6 or more for males and 5 or more for females) [28]. Males reported more frequent episodes of binge drinking [χ2 (3, *n* = 1488) = 44.89, *p* = <0.0005, Cramer’s V = 0.17]. Over half of male drinkers (52.6%, *n* = 377) reported engaging in this behaviour on at least a monthly basis in comparison to 41.4% (*n* = 319) of females. In contrast, the proportion of females who reported that they never binge drink was almost twice that of males (21.7% vs. 10.7%). 

### 3.6. Tobacco Smoking and Illicit Substances

Tobacco smoking (regular or occasional) was less prevalent among males (24.0% vs. 26.8%), although this did not reach statistical significance [χ2 (1, *n* = 1607) = 1.45, *p* = 0.23, Phi = −0.03]. Over half of respondents 57.5% (*n* = 906) reported using marijuana/cannabis at least once in their lifetime, with 38.6% (*n* = 587) using in the previous 12 months of the survey and 17.6% (*n* = 266) in the previous 30 days. There was a significant association between lifetime use and sex, with males generally reporting greater use [χ2 (6, *n* = 1577) = 38.97, *p* = <0.0005, Cramer’s V = 0.16]. Notably, the proportion of males who had used marijuana/cannabis 40 times or more was almost twice the respective proportion of females (21.1% vs. 11.5%).

With regard to lifetime use of other illicit substances, the most prevalent recreational drug among both sexes was Ecstasy/MDMA (males: 22.9%, *n* = 177; females: 12.8%, *n* = 106) followed by Cocaine (males: 19.9%, *n* = 153, females: 10.1%, *n* = 84). A significantly greater proportion of males reported lifetime use of Amphetamine (12.6% vs. 5.9%), LSD (8.9% vs. 4.0%), Cocaine (19.8% vs. 10.1%), Ecstasy/MDMA (22.9% vs. 12.8%), Solvents (3.0% vs. 1.3%), Magic Mushrooms (14.0%vs. 6.8%) and Head Shop Products (11.5% vs. 6.4%) as outlined in Figure 3 below. 

### 3.7. Sexual Health and Behaviours

The vast majority of students (87.7%, *n* = 1348) had been sexually active in the past. Almost 91% classified themselves as ‘heterosexual’ (*n* = 1412), 3.3% ‘homosexual’ (*n* = 52), 4.4% ‘bisexual‘(*n* = 68), 1% ‘asexual’ (*n* = 15) and a small proportion positioned themselves in the ‘other’ category. The median age of first intercourse was 17.0 years with no significant difference between males (Md. 17.0, *n* = 658) and females (Md. 17.0, *n* = 687) [U = 222177.0 z = −0.55, *p* = 0.58, r = −0.01].

Reported lifetime number of sexual partners ranged from 0 to 350. Males reported a statistically greater number of partners than females (Md. 5.0 vs. 4.0), U = 181787.0, z = −4.3, *p* < 0.0005, r = −0.12. Just over 6% of participants (*n* = 85) had been previously told by a doctor that they had a sexually transmitted infection (STI) and there was no significant difference between males and females in this regard [χ2 (1, *n* = 1345) = 2.2, *p* = 0.14, phi 0.04].

A significantly greater proportion of females cited having intercourse with one partner as a protective mechanism against STIs (56.1% vs. 44.0%) and expected their partner to have an STI test (8.0 vs. 3.6%). Males were more likely to have used alcohol/drugs prior to their most recent intercourse (34.8% vs. 28.5%, *p* = 0.02) as outlined in Table 4.

### 3.8. Sleep 

With regard to recent sleep quality (*n* = 1543), 29.0% rated this parameter as either ‘very poor’ (6.6%, *n* = 102) or ‘poor’ (22.4%, *n* = 345), 27.8% as ‘neither poor nor good’, 30.6% as ‘good’ and 12.6% as ‘very good’. With regard to duration, 79.3% (*n* = 1215) did not meet the recommended 8-hour threshold during the week (Mon to Fri). There was no significant difference between males and females in terms of perceived sleep quality or reported sleep duration.

### 3.9. Mental Wellbeing

In total, 12.6% (*n* = 194) rated their overall mental health as either ‘very poor’ (2.4%) or ‘poor’ (10.2%), 22.6% (*n* = 347) as ‘neither poor nor good’, 42.6% (*n* = 654) as ‘good’ and 22.1% (*n* = 339) as ‘very good’. Males generally rated their mental health more favourably than females [χ2 (4, *n* = 1534) = 25.97, *p* = <0.0005, Cramer’s V = 0.13]. Of note, a greater proportion of males rated their health as ‘very good’ versus females (27.2% vs. 17.3%). Males also scored significantly higher than females (and therefore more favourably from a mental well-being perspective) on both the Energy & Vitality Index (EVI): [Md. 50.0, *n* = 744 vs. Md. 40.0, *n* = 788), U = 22064705.0, z = −8.4, *p* < 0.0005, r = −0.21] and the Mental Health Index-5 (MHI-5): [Md. 68.0, *n* = 742 vs. Md. 60.0, *n* = 789, U = 224799.0, z = −7.87, *p* < 0.0005, r = 0.20]. Table 5 outlines the proportion of male and female students that reported being ‘highly stressed’ by each respective stressor. A greater proportion of females were highly stressed by all of the listed stressors. Mann Whitney U tests revealed that, for each stressor, MHI-5 scores were significantly lower (representing greater negative mental health symptoms) among students who reported being ‘highly stressed’ versus those who did not. 

### 3.10. Social Media 

Only 3.6% of total participants (*n* = 56) did not have a social media account of any kind. The most common platforms reported were Facebook (93.0%, *n* = 1430), Snapchat (69.2%, *n* = 1064) and Instagram (54.2%, *n* = 833). Females were more likely to have a Facebook, Instagram and Snapchat account and males were more likely not to have an active social media account at all (Table 6). 

Females were more likely to have posted pictures than males during the 30 days prior to the survey (73.7% vs. 57.2%), [χ2 (1, *n* = 1462) = 43.55, *p* =< 0.0005, Phi = 0.17]. Of those who posted pictures (65.8% of applicable participants), 44.0% (*n* = 425) posted pictures of themselves either drinking or being around alcohol. During ‘weekdays’ (Mon-Fri), 31.6% (*n* = 464) spent 90 min or more per day on social media and there was no significant difference between males and females in this regard (30.3% vs. 32.8%), [χ2 (1, *n* = 1469) = 0.99, *p* = 0.32, phi = −0.03]. A significantly greater proportion of females reported spending 90 min or more on social media per day at the weekend (35.1% vs. 27.0%) [χ2 (1, *n* = 1457) =10.66, *p* = 0.001, phi = −0.08].

### 3.11. Predictors of Energy and Vitality Index (EVI) Scores

Multivariate linear regression was carried out with EVI score as the dependant variable. Independent categorical variables were dichotomised as outlined below. Based on findings outlined above, initial independent variables included as predictors were sex (0 = male,1 = female), age, having a ‘good’ or ‘very good’ general health perception (dichotomised as 0 = no, 1 = yes), habitual daily fruit and vegetable servings, adjusted physical activity volume (moderate and vigorous only, as described above), AUDIT-C scores, tobacco smoking (0 = no 1 = yes), lifetime cannabis/marijuana use (0 = never or less than 20 times, 1 = at least 20 times or more), ‘good’ or ‘very good’ sleep quality (0 = no, 1 = yes), ‘good’ or ‘very good’ mental health perception (0 = no, 1 = yes) using social media for at least 90 min per day Mon-Fri (0 = no or not applicable, 1 = yes), lifetime use of ecstasy (0 = no,1 = yes), calculated BMI, self-perceived weight category (0=underweight or normal weight 1=overweight/obese), sitting time (0 = up to 4 h, 1 = 4 h or more) and number of sexual partners. In terms of missing data points due to survey skip logic, non-drinkers (*n* = 182) were included as having an AUDIT-C score of ‘0’, those not sexually active (*n* = 189) included as having ‘0’ sexual partners, and those without an active account (*n* = 56) included as spending ‘0’ daily minutes on social media. Only the independent variables that displayed a statistically significant influence in the initial model were included in the final model (Table 7). This model had an adjusted R-squared value of 0.370, thus accounting for 37.0% of the variance in EVI scores. 

## 4. Discussion

This study provides an insight into the lifestyle behaviours of students at a Higher Education Institution (HEI) in southern Ireland. In the most recent academic year, there were 231,710 full and part-time enrolments in Irish HEIs, a figure that has consistently increased over the past five years [64]. Therefore, in a time where non-communicable diseases constitute an unprecedented public health challenge, habitual behaviours of this growing cohort are of cross-sectoral interest from the perspective of preserving the health of generations to come [65]. 

All registered students of the Institute (*n* = 11,261) with a valid student email address were eligible, and provided with an opportunity to participate. This approach was deemed preferable to probabilistic sampling given that the fundamental purpose of this research was to inform an institute-wide campus health promotion initiative (*A Healthy CIT*). From this perspective, it was essential that all students were provided with an equal opportunity to contribute. Secondly, in-classroom response bias was a concern, due to the smaller class numbers at Cork Institute of Technology relative to many other HEIs. Dissemination via email afforded students the opportunity complete the questionnaire in a more private setting, given the sensitive nature of certain items. 

In the absence of comparative literature, the initial response rate (21.2%) could be perceived as low. However, in terms of the absolute value, the reported sample size was greater than previously-reported values from a multitude of single-HEI studies. In an Irish context, a previous study carried out at National University of Ireland Galway [52] reported a sample size of 841. In addition, although a contrasting probabilistic sampling method was employed, a study conducted at University College Cork reported a sample size of 2275 [28]. It should be borne in mind that, in terms of student enrolments, both of these aforementioned HEIs are of a greater magnitude than Cork Institute of Technology. Internationally, in a study of students attending the University of Marburg in Germany [21], a total of 1319 initial responses were received. This study adopted a convenience sampling approach and was logistically restricted to three of the University’s Schools [21]. Similarly, convenience sampling was employed in a study conducted at a single UK university where 410 responses to a health and lifestyle questionnaire instrument were received, representing 16% of the HEI sample [47]. Finally, the current study’s response rate is also marginally lower than the response rate (27%, *n* = 985) received to a health questionnaire confined to a single campus (Pontevedra) of the University of Vigo in Spain [49]. When contextualised in terms of previous Irish and international work, the sample size of the current study appears to compare favourably in this regard. 

Although over three-quarters (78.6%, *n* = 1667) of participants rated their health as either ‘very good’ or ‘good’; this is lower than the most recent age-stratified comparative figures from the general Irish population where 93% of 15–24 year olds and 94% of 25–34 year olds rated their general health within these parameters [66]. General health was rated more favourably by males in the current study, which is consistent with previous Irish research [24] as well as an international study involving first year undergraduate students in Sweden [67].

The calculated prevalence of overweight/obesity for male (*n* = 979) and female (*n* = 1011) participants (41.9% and 34.7% respectively) was greater than previously reported values of 17% (males) and 6.1% (females) within a cohort of university students from 22 countries across Europe, North America, Asia and South America [68]. That particular study also highlighted a tendency for students to over-report height and under-report body mass, hence potentially underestimating their overall BMI value [68]. Due to a similar reliance on self-reported data in the current study, the true prevalence of overweight and obesity may be even greater in the present cohort. Given the established association between overweight/obesity and increased relative risk of all-cause mortality on a global scale [69], this is a concerning trend that, if maintained, could result in a plethora of detrimental sequelae in later life [70].

The discrepancy between students’ self-perceived and calculated BMI category is also a concern. Although the Kappa measure of agreement, a commonly used indicator of inter-rater reliability, reached statistical significance for these variables (Kappa 0.41, *p* < 0.0005), in practical terms this may constitute a weak level of agreement according to revised interpretations of this statistic [71]. Level of agreement was lower for males than females (0.32 vs. 0.50), which is consistent with findings in a Canadian study [72] but contradictory to findings in a study involving over 34,000 employees of a multi-centre financial company in Japan [73].

Among males, ‘underweight’ was overestimated by 188%, ‘normal weight’ overestimated by 24%, ‘overweight’ underestimated by 29% and ‘obesity’ underestimated by 83%. This suggests that, for males, the magnitude of the discrepancy between perceived and calculated BMI is greatest at either very low (underweight) or very high (obese) BMI values and less exaggerated within the normal and overweight categories.

Qualitatively, the following trends were observed among those who provided height/body mass data as well as their self-perceived weight category. Firstly, obesity was underestimated by both females (4% perceived prevalence vs. 11.3% calculated prevalence) and males (1.8% perceived vs. 10.7% calculated). However, a greater proportion of females perceived themselves to be overweight relative to the calculated overweight prevalence (30.7% vs. 23.0%). Conversely, only 22.4% of males perceived themselves as overweight whereas mathematical calculations classified a greater proportion of 31.4% within this category. 

Strong evidence has accumulated to suggest that a significant proportion of those who are overweight or obese either perceive themselves to be of normal weight or fail to recognise the extent of their overweight/obesity due to the ‘visual normalisation’ of larger body types in contemporary society [74]. The findings of the current study align with this trend for both sexes with respect to obesity but only for males regarding overweight. The fact that a greater proportion of females classified themselves as ‘overweight’ relative to the calculated prevalence could be attributable to the attenuating effect of visual exposure to aesthetically lean body types in popular media targeted towards female cohorts [74]. The tendency of normal or underweight females to categorise themselves as overweight is also consistent with previous research involving university students across seven European countries [16]. This study concluded that, despite concurrent self-reported height/body mass values yielding a normal calculated BMI, females tended to categorise themselves as excessively fat and males perceived themselves as excessively thin [16]. Future mixed-method studies are required to explore sex-specific patterns of discrepant anthropometric perceptions, to determine the reasons for such discrepancies and determine the extent to which the aforementioned ‘visual normalisation’ of overweight/obesity [74] influences perceptions. Quantitatively, such research should directly compare self-reported, objectively measured and self-perceived BMI within the same population. Qualitatively, potential social mediators of body image perceptions need to be explored. 

From a nutritional perspective, daily fruit and vegetable intake was sub-optimal (median 3.0 servings). In Ireland, the Department of Health has recently revised its dietary guidelines for the population, increasing the recommended daily servings of fruit and vegetables to 7 per day [75]. In the current study, only 5.5% of males and 5.7% of females habitually achieved this. In terms of absolute values, females reported more daily servings than males, a trend that is consistent with extant literature [7,21,47]. 

Conversely, a greater proportion of males attained the minimum recommended volume of physical activity during the 7 days prior to respective survey completion (74.8% vs. 67.8%, *p* = 0.003). This substantiates the findings of previous studies that reported a tendency for male students to be more physically active than their female counterparts [24,48,49].

Sitting time was ingrained within the college day, coupled with a perceived lack of time reported as the predominant barrier to physical activity by both sexes. A recent meta-analysis reported that, regardless of physical activity, the risk of death from all causes increased by 2% for each hour of sitting time per day, with further increases of 5% for each additional hour of sitting time at or greater than a total of 7 h per day [76]. HEIs should give consideration to this issue at a macro level whereby the cultural norms of an institution should not intrinsically promote sedentary behaviours. Academic curricula should incorporate, rather than impede, opportunities for students to become more physically active. 

The reliability of the AUDIT instrument has been confirmed in a variety of settings [77,78], including a HEI cohort in the USA [79]. The instrument and/or its sub-domains have been utilised internationally to screen for hazardous drinking in HEI student populations [28,29,31,32,33,36,80]. Although hazardous drinking classifications were comparable between sexes (54.1% of females, 54.7% of males), males scored significantly higher on the AUDIT instrument and engaged in binge drinking more frequently. From a risk stratification perspective, a greater proportion of males (34.8% vs. 28.5%, *p* = 0.02) consumed alcohol and/or used drugs prior to their most recent sexual intercourse. Whilst this relationship requires further investigation, it suggests that male students adopt a riskier attitude when under the influence of alcohol and/or drugs, which could result in impaired decision-making that may impact related lifestyle domains (sexual health for example). In terms of sexual risk-taking, males reported a greater number of lifetime sexual partners and were less likely to expect their partner to take an STI test. Males were more likely to have used a number of recreational substances, including Ecstasy/MDMA (22.9% vs. 12.8%) and Cocaine (19.9% vs. 10.1%) which is consistent with previous literature [24].

Females experienced greater levels of psychological stress and also achieved statistically lower scores (and therefore less favourable) in both the Energy and Vitality Index (EVI) (positive mental health characteristics) and the Mental Health Index- 5 (negative mental health symptoms) measures. This mirrors trends in the general Irish population [81] as well as previously reported trends among HEI students in Ireland [52,82].

Of note, mean MHI-5 scores were lower relative to the general Irish population [81] for both females (57.9 vs. 81.8) and males (65.7 vs. 85.5). This was also the case with EVI mean scores for both sexes (females: 41.8 vs. 65.9, males: 50.5 vs. 69.8). Although a contrasting measurement scale was utilised, this trend has been mirrored internationally in Australian research that reported an 83.9% prevalence of psychological distress among a sample of 6479 university students relative to a prevalence of 29% among the general Australian population [83]. Furthermore, research involving 5572 university students from 12 countries across Europe, Asia and North America highlighted a 28.8% prevalence of suicidal thoughts as well as a 33.6% prevalence of clinically-relevant psychological distress [84]. Therefore, it is evident that although certain stressors are interwoven within the higher education experience, institutions need to be aware of the significant and measurable detrimental effect that such stressors may be exerting on students’ mental health.

Of all self-rated parameters (general health, recent sleep quality, mental health), sleep quality was rated most negatively (29.0% rated this as either ‘very poor’ or ‘poor’). A substantial body of evidence has accrued in recent years demonstrating an inverse relationship between sub-optimal sleep and academic performance at third level [85,86,87,88,89,90]. Only one-fifth of students (20.7%, *n* = 318) reported getting at least 8 h sleep on a typical weeknight (Mon-Fri), but a tendency to compensate was observed at weekends with an increased value of 58.3%. This may be due to later bed times and waking times at the weekend, which should be addressed in order to stabilise circadian rhythms and standardise sleep patterns across the week [91]. 

Although the vast majority of students (96.4%) had at least one active social media account, males were more likely not to have any (5.5% vs. 1.9%, *p* < 0.0005). Notably, from Monday to Friday, almost one third who had any active account (31.6%) spent at least 90 min or more each day on social media. Using social media while studying could be associated with inadequate time management and disproportionate cramming prior to examinations, as was found in previous Irish research involving a cohort of medical students [62]. 

The second objective was to identify the significant health and lifestyle predictors of positive mental health, using EVI scores as the dependent variable. Sex was included as an independent variable given the fact that statistically significant differences in EVI scores were revealed by a Mann Whitney U Test as outlined above. A regression equation was derived that accounted for 37.0% of the variance in EVI scores with no evidence of multi-collinearity between independent variables and a straight line relationship between residuals and predicted EVI scores.

Positive predictors were having a ‘good/very good’ general health perception, habitual daily fruit and vegetable servings, AUDIT-C scores (items 1–3 of total AUDIT instrument) ‘good/very good’ mental health rating and ‘good/very good’ recent sleep quality. Negative predictors were female sex, spending at least 90 min per day on social media Mon-Fri (0 = no, 1 = yes), and having a perception of being overweight/obese (0 = underweight/normal weight, 1 = overweight/obese). 

It is not surprising that the predictors with the greatest influence on EVI scores were having a positive (good/very good) perception of mental health (standardised β 0.32), recent sleep quality (standardised β 0.27) and general health (standardised β 0.15). The standardised beta value obtained for AUDIT-C scores however (0.05) suggests that as scores on this subscale measure of hazardous drinking increase, a positive (and therefore more favourable) influence on EVI score is exerted. This relationship could be explained by the fact that alcohol consumption is heavily ingrained within Irish society [92] and appears inextricably linked with the higher education lifestyle [93]. Students who consume alcohol as part of social activities may therefore feel a greater sense of social acceptance and this in turn could enhance their positive mental health. This hypothesis is supported by the work of Capron et al. [80] who acknowledged that for certain students, the immediate benefits, such as social lubrication, may be sought after despite the later cost of negative consequences. A qualitative exploration of this finding would be particularly beneficial to guide the design of cohort-specific alcohol interventions, alternative alcohol-free social activities and institutional policies going forward. 

It is notable that calculated BMI values were not a significant predictor of EVI scores in the initial model, but having a perception of being overweight or obese was a statistically significant negative predictor included in the final model. This should be interpreted in the context of the relatively weak level of agreement between self-perceived and calculated BMI category. In the first instance, there is a requirement to investigate and address the evident discrepancies between calculated BMI versus subjectively perceived, and socially influenced, attitudes towards body habitus. The findings of the current paper provide a definitive rationale for this work given that inaccurate perceptions of overweight/obesity, particularly among female students, may be negatively impacting mental health.

### 4.1. Strengths

The predominant strength is the incorporation of a multitude of health and lifestyle domains within a single study hence facilitating multivariate analysis. Validated scales will facilitate future longitudinal studies as well as a myriad of international comparisons within each health domain. This work also serves to address the paucity of contemporary data pertaining to the lifestyle behaviours of HEI students in Ireland. Pragmatically, this study provides empirical evidence to assist HEIs in terms of resource allocation and strategic prioritisation of health-related domains in light of ongoing funding constraints [51]. Furthermore, at the time of writing, planning of the second phase of *A Healthy CIT’s* programme of research activities has commenced. This phase will involve dissemination of a second iteration of the questionnaire instrument outlined above to enable retrospective comparisons with baseline data described in the current paper.

### 4.2. Limitations

Due to the cross-sectional design employed, associations cannot be deemed causal. Secondly, although all students were eligible to participate, a convenience sample was employed, which potentially introduced a selection bias. Therefore, findings may not be representative of all students of the Institute. As there was an inevitable reliance on self-report data, under/over-reporting, egoism and/or socially desirable responses cannot be ruled out. It is also possible that there was variability in reported lifestyle trends and health metrics between participants who responded to the survey during the initial data collection period (prior to examinations) versus those who responded during the subsequent 7-day period (post examinations). However, any alleged variability should be considered in the context of the semesterised nature of programme delivery at Cork Institute of Technology. An inherent culture of continuous assessment exists within the Institute in order to disperse student workloads more evenly throughout the academic year. Therefore, it is less likely that the end of semester formal examinations would have significantly influenced habitual lifestyle patterns. Finally, the findings of this study may not be entirely generalisable to Irish or international student cohorts, due to inherent cultural differences that exist between HEIs.

## 5. Conclusions

Health and lifestyle behaviours among this cohort were sub-optimal and would benefit from sex-specific multi-level initiatives incorporating institutional policy review. Initial concerns are hazardous drinking behaviours, sub-optimal fruit and vegetable intake, poor sleep quality and stark misperceptions of anthropometric status. Males may adopt riskier attitudes towards alcohol, illicit substances and sexual behaviours, despite more favourable perceptions of general health. Females appear to exhibit greater levels of psychological stress, particularly with regard to academic workload. Future longitudinal studies are required to make inferences regarding causality. Qualitative exploration would be particularly worthwhile to gain a greater understanding of the micro and macro-level mediators of health-related behaviours during this formative stage. 

## Figures and Tables

**Figure 1 ijerph-16-04318-f001:**
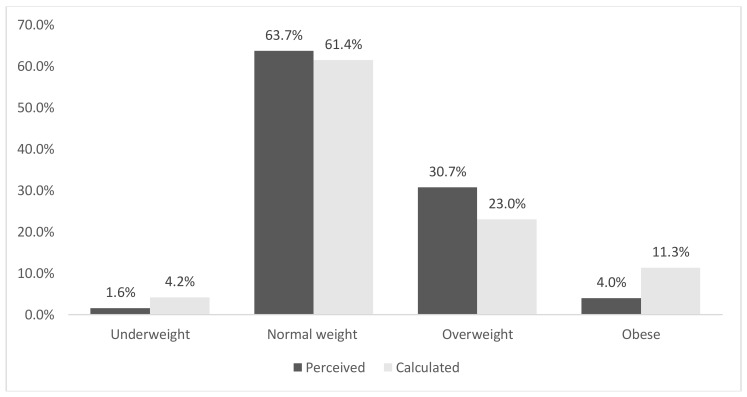
Perceived versus calculated BMI category classifications (females, *n* = 996). Kappa Measure of Agreement 0.50, *p* < 0.0005.

**Figure 2 ijerph-16-04318-f002:**
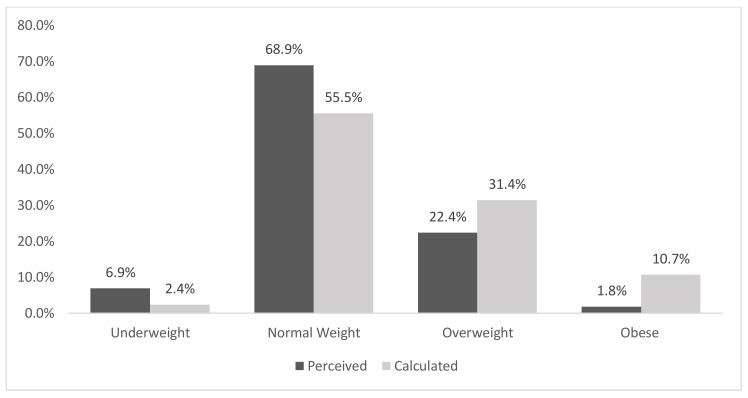
Perceived versus calculated BMI category classifications (males, *n* = 974). Kappa Measure of Agreement 0.32, *p* < 0.0005.

**Figure 3 ijerph-16-04318-f003:**
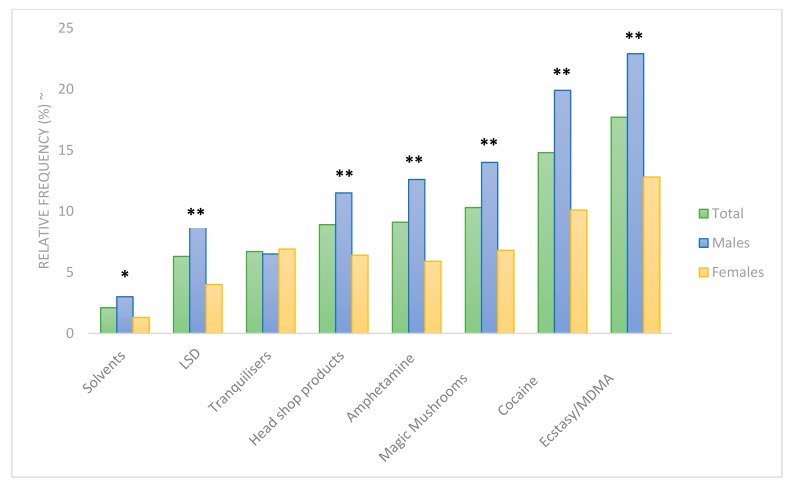
Lifetime use of illicit substances by sex. *significant between sexes *p* < 0.05 **significant between sexes *p* < 0.0005, ~Item valid responses: Solvents *n* = 1597, LSD *n* = 1595, Tranquilisers *n* = 1602, Head Shop Products *n* = 1598, Amphetamine *n* = 1599, Magic Mushrooms *n* = 1598, Cocaine *n* = 1600, Ecstasy/MDMA *n* = 1601. Due to low absolute numbers reporting their use, relevin, heroin and drugs by injection were omitted from this analysis.

**Table 1 ijerph-16-04318-t001:** Demographic Characteristics of Participants by Sex.

Demographic Variable	Total		M		F		*p*-Value *
	*n*	%	*n*	%	*n*	%	
Sex	2267		1094	48.3	1173	51.7	
Age							
18–20 years #	806	35.6	357	32.7	449	38.4	0.003
21–23 years	754	33.3	362	33.1	392	33.5	
24 years and older #	703	31.1	374	34.2	329	28.1	
Total	2263		1093		1170		
Mode of study							
Undergraduate	1891	90.2	907	89.4	984	91.0	0.22
Postgraduate (Taught)	133	6.3	66	6.5	67	6.2	
Postgraduate (Research)	72	3.4	42	4.1	30	2.8	
Total	2096		1015		1081		
Registration status							
Full-time	1918	88.4	919	87.6	999	89.2	0.28
Part-time or evening	251	11.6	130	12.4	121	10.8	
Total	2169		1049		1120		
Faculty ^**^							
Business & Humanities #	941	42.9	327	30.8	614	54.2	<0.0005
Engineering & Science #	922	42.0	579	54.6	343	30.3	
Crawford College of Art & Design #	169	7.7	59	5.6	110	9.7	
Cork School of Music #	98	4.5	37	3.5	61	5.4	
National Maritime College of Ireland #	51	2.3	49	4.6	2	0.2	
Other #	13	0.6	10	0.9	3	0.3	
Total	2194		1061		1133		
Year Group (undergraduates)							
1	642	32.6	298	31.7	344	33.5	0.49
2	510	25.9	256	27.2	254	24.7	
3	464	23.6	213	22.7	251	24.4	
4	338	17.2	165	17.6	173	16.8	
Other	13	0.7	8	0.9	5	0.5	
Total	1967		940		1027		
Year Group (postgraduates)							
1	97	47.5	47	43.5	50	52.1	0.06
2	49	24.0	22	20.4	27	28.1	
3	20	9.8	11	10.2	9	9.4	
4	23	11.3	16	14.8	7	7.3	
5/6/Other #	15	7.4	12	11.1	3	3.1	
Total	204		108		96		
Nationality							
Irish #	1937	89.2	954	91.0	983	87.5	
UK	30	1.4	16	1.5	14	1.2	0.006
Other European #	141	6.5	48	4.6	93	8.3	
Other	64	2.9	30	2.9	34	3.0	
Total	2172		1048		1124		
Academic Achievement (undergraduates only)							
Less than 40%	11	0.6	6	0.6	5	0.5	0.02
40–59% #	464	23.9	253	27.2	211	20.9	
60–69%	766	39.5	357	38.3	409	40.6	
70% or above	491	25.3	226	24.3	265	26.3	
Don’t Know #	144	7.4	57	6.1	87	8.6	
Declined to Provide this Information	63	3.2	32	3.4	31	3.1	
Total	1940		931		1008		

* Chi Squared Test for Independence across all response categories (continuity correction applied for 2 × 2 tables), ~ collapsed continuous age variable, ** recoded based on students’ reported area of study, # significant within-variable category differences between male/female proportions based on multiple comparisons with Bonferroni adjustment to *p*-value.

**Table 2 ijerph-16-04318-t002:** Perceived Barriers to Engaging in More Physical Activity and Daily Sitting Time by Sex.

Physical Activity Variable	Total		M		F		*p*-Value *
	*n*	%	*n*	%	*n*	%	
Perceived Barriers							
No time/workload/exams	862	51.6	408	51.1	454	52.2	<0.0005
Already take enough exercise #	208	12.5	122	15.3	86	9.9	
Interested but not willing to spend time	170	10.2	85	10.6	85	9.8	
Injury/disability/medical # condition/pregnancy	124	7.4	48	6.0	76	8.7	
Not the sporty type	93	5.6	36	4.5	57	6.6	
No access to facilities #	65	3.9	21	2.6	44	5.1	
Not interested	63	3.8	30	3.8	33	3.8	
Exercising too much/overtraining #	43	2.6	32	4.0	11	1.3	
Other	41	2.5	17	2.1	24	2.8	
Total	1669		799		870		
Sitting time during average day at college (hours)							
Less than 1 hour	25	1.4	8	0.9	17	1.9	0.48
1–2 h	56	3.2	26	3.1	30	3.4	
2–3 h	163	9.4	76	9.0	87	9.8	
3–4 h	351	20.2	180	21.3	171	19.3	
4–5 h	475	27.4	227	26.8	248	28.0	
More than 5 h	664	38.3	330	39.0	334	37.7	
Total	1734		847		887		

* Chi Squared Test for Independence across all response categories, # significant within-variable category differences between male/female proportions based on multiple comparisons with Bonferroni adjustment to *p*-value.

**Table 3 ijerph-16-04318-t003:** Alcohol Use Disorders Identification Test (AUDIT) and sub-domain scores by sex.

AUDIT Domain	Items	Cronbach’s α	Total		Males		Females		*p*-Value #
			*n* *	Median	*n*	Median	*n*	Median	
Total AUDIT	1–10	0.82	1329	7.0	651	8.0	678	7.0	<0.0005
Hazardous Alcohol Use	1–3	0.66	1469	5.0	707	6.0	762	5.0	<0.0005
Dependence symptoms	4–6	0.61	1430	0.0	694	0.0	736	0.0	0.57
Harmful alcohol use	7–10	0.68	1412	2.0	683	2.00	729	1.0	0.43

* Data presented excludes non-drinkers (*n* = 182) who were not exposed to further items having stated they never drink in item 1. # Mann Whitney U Test.

**Table 4 ijerph-16-04318-t004:** Sexual health behaviours: males versus females.

‘In What Way Do You Protect Yourself from a Sexually Transmitted Infection?’ (Yes)	Total (*n* = 1356)~	Males (*n* = 659)	Females (*n* = 697)	*p*-Value *
I don’t protect myself at all	6.2	6.8	5.6	0.41
I protect myself by use of a condom	61.6	68.3	55.2	<0.0005
I have intercourse with only one constant partner	50.2	44.0	56.1	<0.0005
I expect my partner to have an STI test	5.9	3.6	8.0	0.001
Other	1.4	0.8	2.0	0.08
**‘Did You Drink Alcohol and/or Use Drugs before You Had Sexual Intercourse the Last Time?’ (Yes/No)**	**Total** **(*n* = 1331)**	**Males** **(*n* = 650)**	**Females** **(*n* = 681)**	***p*** **-Value ***
Yes	31.6	34.8	28.5	0.02
**‘Have You (or Your Partner) ever Used the Morning after Pill?’ (Yes/No)**	**Total** **(*n* = 1322)**	**Males** **(*n* = 629)**	**Females** **(*n* = 693)**	***p*** **-Value ***
Yes	44.4	36.7	51.4	<0.0005

All figures are percentage values. * Chi Squared Test for Independence (2 × 2 cross tabulation male/female*yes/no).

**Table 5 ijerph-16-04318-t005:** Psychological stressors (‘highly stressed’) by sex and differences in MHI-5 scores between groups.

Psychological Stressor (‘Highly Stressed’)	Total		Males		Females		Md. MHI-5 Scores (‘Highly Stressed’ vs. other Categories) #	
	(*n*)	%	(*n*)	%	(*n*)	%		
Exams	524	34.6	179	24.2 ^***^	345	44.5	56.0 ^***^	68.0
College workload	432	28.3	149	20.1 ^***^	283	36.1	56.0 ^***^	68.0
Studies in general	377	24.6	122	16.4 ^***^	255	32.4	50.0 ^***^	68.0
Financial situation	352	23.1	138	18.6 ^***^	214	27.2	53.5 ^***^	68.0
Pressure outside of work/college	290	19.4	93	12.8 ^***^	197	25.7	52.0 ^***^	68.0
Competition at college	160	10.7	52	7.1 ^***^	108	14.0	48.0 ^***^	64.0
Living situation	147	9.7	46	6.2 ^***^	101	13.0	44.0 ^***^	64.0
Family situation	141	9.3	45	6.1 ^***^	96	12.3	44.0 ^***^	64.0
Relationships/sexuality	98	6.5	38	5.2	60	7.7	40.0^***^	64.0
Illness	68	4.5	17	2.3 ^***^	51	6.6	32.0 ^***^	64.0
Circle of friends	55	3.6	18	2.5 ^*^	37	4.7	32.0 ^***^	64.0
Social Media	33	2.2	9	1.2 ^*^	24	3.1	36.0 ^***^	64.0

*** *p* < 0.0005, ** *p* < 0.01, * *p* < 0.05. # ’highly stressed’ vs. those who selected one of the other categories in the Likert scale (‘often stressed’, ‘not often stressed’ or ‘never stressed’). ‘Other’ was excluded from this analysis (*n* = 20).

**Table 6 ijerph-16-04318-t006:** Social media platforms by sex.

Social Media Platform	Total (*n* = 1537)	Males (*n* = 751)	Females (*n* = 786)	*p*-Value
Facebook	93.0	90.3	95.7	<0.0005
Snapchat	69.2	64.2	74.0	<0.0005
Instagram	54.2	42.7	65.1	<0.0005
Twitter	40.6	40.6	40.6	1.0
LinkedIn	27.4	29.2	25.7	0.14
No social media account	3.6	5.5	1.9	<0.0005
Other	2.7	2.5	2.9	0.75

All figures are percentage values.

**Table 7 ijerph-16-04318-t007:** Multiple Linear Regression Model with EVI Scores as the Dependent Variable.

Predictor Variable	Standardised β	*p*-Value
Sex	−0.15	<0.0005
General health rating	0.15	<0.0005
Daily total fruit and vegetable servings	0.08	<0.0005
AUDIT-C scores	0.05	0.04
Mental health rating	0.32	<0.0005
Recent sleep quality	0.27	<0.0005
Social media 90 min or more on weekdays	−0.06	0.01
Perception of being overweight/obese	−0.09	<0.0005
Adjusted R Squared of the Model	0.370	<0.0005

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
