# Peer review of "‘A Healthy CIT’: An Investigation into Student Health Metrics, Lifestyle Behaviours and the Predictors of Positive Mental Health in an Irish Higher Education Setting"

_ijerph, 2019, doi:10.3390/ijerph16224318_

Round 1

Reviewer 1 Report

This was a computer survey research using online questionnaire. The research subjects were all registered students at Cork Institute of Technology(n=11,261) in Semester Two (Spring) of a standard academic year (2016?). The online questionnaire contained 92 main items that seemed complex.

Because the questionnaire with 92 main items  was too long, it may be the reason why only 2,390 responses (21.2% response rate), and  furthermore, removing missing and spoiled data, only 20.1% total response rate.

In addition, this was self-report online survey. The answers of college students may skew to self-interest (egoism), thus I think the baseline data analysis would be unworthy.

I  will suggest use of dataset from ‘A Healthy CIT’ waiting for 2 or 3 years latter. The authors could think about a smart design to overcome the short response rate and the problem of students egoism. 

Author Response

Response to Reviewer 1 Comments

Point 1: This was a computer survey research using online questionnaire. The research subjects were all registered students at Cork Institute of Technology(n=11,261) in Semester Two (Spring) of a standard academic year (2016?). The online questionnaire contained 92 main items that seemed complex

Response 1: The Authors thank the Reviewer for Point 1. We confirm that the data was collected during the Spring Semester of a standard academic year (2016).

The Authors appreciate the Reviewer’s perception regarding the alleged complexity of the instrument. However, we believe that there was a strong rationale for adopting this approach. Previous work has involved the use of multi-faceted survey instruments and the Authors refer the Reviewer specifically to two such papers that have been published in the current journal (Varela-Mato et al., 2012; El Ansari et al., 2011).

Furthermore, as identified by the Reviewer in Point 1, the survey was disseminated online. Online surveys have been reported to reduce participant burden relative to traditional paper based instruments (Kwak & Radler, 2002). Our survey was hosted on a user-friendly, institute branded interface that would have been acceptable and easy to follow by the target cohort. In addition, although the instrument contained a maximum of 92 main items, skip logic was embedded within the platform to ensure that participants were exposed only to items that were of specific relevance to them.

In order to further minimise complexity and increase acceptability in terms of nomenclature, and structure, a previous iteration of the instrument (in paper form) was piloted as part of a classroom activity with a cohort of undergraduate students (n=13). Revisions were subsequently made to the instrument based upon this feedback. In addition, as described in section 2.2 of the manuscript (Lines 131 to 246) many items were sourced directly from  previous research in Ireland involving a cohort of undergraduate students from 21 Higher Education Institutions (HEIs) (Hope et al., 2005). These items have therefore demonstrated previous validity in a comparative cohort. The reliability of further scales within the instrument has also been demonstrated. For example, the Alcohol Use Disorders Identification Test and its subscales have been used extensively to evaluate drinking behaviours among higher education student cohorts worldwide (Davoren et al., 2015; Kokotailo et al., 2004).

In summary, the Authors believe that the content of the questionnaire instrument was presented in an appropriate, logic-enabled format that reduced its complexity and maximised its validity. Rather than viewing the comprehensive nature of the survey as a limitation, the Authors refer the Reviewer to the work of Dodd and colleagues (2010), who concluded that ‘unhealthy behaviours do not occur in isolation. Interventions should consider how behaviours cluster/interact together and design effective programmes that mirror the patterns within clusters’. As stated in Lines 122-123, the current study forms part of the broader designated research activities of our institute’s new campus health promotion initiative (A Healthy CIT). The lack of empirical data guiding such initiatives within Irish HEIs impedes any rigorous evaluative efforts and the extent to which multi-faceted interventions can be tailored appropriately. The Authors believe that the inclusion of a multitude of health and lifestyle domains in our questionnaire instrument has served to address this dearth of data and will be of interest to policy-makers and practitioners in similar institutions elsewhere.

Point 2: Because the questionnaire with 92 main items  was too long, it may be the reason why only 2,390 responses (21.2% response rate), and  furthermore, removing missing and spoiled data, only 20.1% total response rate.

Response 2: The Authors thank the Reviewer for this observation. We once again refer the Reviewer to the rationale for the comprehensive nature of the instrument as outlined in our response to Point 1 above.

In addition, the Authors understand that, in the absence of comparative literature, the response rate may have appeared low to the Reviewer. However, it is substantially higher than the 7.3% response rate reported by Perusse-Lachance and colleagues (2010) to an online survey distributed to a cohort of higher education students in Canada. Our response rate is also only marginally lower than the response rate (27%) reported in a previous cross-sectional study involving higher education students in Spain (Varela-Mato et al., 2012).

We note the Reviewer’s comment that there were ‘only 2,390 responses’. In order to provide clarification, we would like to discuss response numbers relative to previous work in the field. In terms of the absolute value, our reported sample size was greater than many comparative single-setting (i.e. a single HEI) studies. In an Irish context, previous studies have reported sample sizes of 841 (a study carried out in National University of Ireland [NUIG], Galway; Mac Neela et al., 2012) and 2,275 (a study carried out at University College Cork [UCC]; Davoren et al., 2015). For comparative purposes, if one considers each HEI’s total student enrolment figures pertaining to the year of data collection (published annually by the Irish Higher Education Authority), the sample sizes listed above would equate to 5.5% of total enrolments at NUIG (n=15,302; 2008/2009) and 12.6% of total enrolments at UCC (n=18,088; 2011/2012). Therefore, the sample size and response rate of the current study (n=2,390, 21.2%) compare favourably in this regard.

From an international perspective sample sizes of 410 (UK; Dodd et al., 2010), 1,656 (Saudi Arabia; Almutairi et al., 2018), 985 (Spain; Varela-Mato et al., 2012) and 535 (USA; Robinson et al., 2016) have also been reported in similar single setting studies. Furthermore, as a seminal reference for any health-related survey involving higher education students in Ireland, the College Lifestyle and Attitudinal Survey (Hope et al., 2005) was a collaborative study involving 21 HEIs in Ireland. This study had a total sample size of 3,259. Similarly, a study involving 7 UK universities reported a sample size of 3,706 (El Ansari et al., 2011).

Finally, missing data is an inherent aspect of any population survey. Once again we refer the Reviewer to the work of Varela-Mato et al. (2012) who reported a considerable proportion of missing data. The primary objective of the current paper was to identify sex related differences in lifestyle behaviours. At the outset, a Chi Squared test for Independence was carried out to ascertain whether sex was related to completing or not completing the questionnaire. As documented in Lines 291-292: ‘There was no significant association between sex and progressing to the final section of the questionnaire instrument [χ2 (1, n=2,267) =0.64, p=0.43 Phi 0.02]’.

Point 3: In addition, this was self-report online survey. The answers of college students may skew to self-interest (egoism), thus I think the baseline data analysis would be unworthy.

Response 3: The Authors thank the Reviewer for Point 3. In any research study involving self-reported data, socially desirable responses, under/over-reporting, recall bias and, as outlined by the Reviewer, ‘egoism’ are widely recognised potential limitations. This is acknowledged specifically in Lines 677-678 of the manuscript (‘As there was an inevitable reliance on self-report data, under/over-reporting and/or socially desirable responses cannot be ruled out’). In the current study, it was explicitly clear to participants (in email communications and on the landing page of the online survey) that the survey was anonymous. This, coupled with the fact that the survey was not classroom or paper-based, would have minimised the incidence of socially desirable responding.

Cross-sectional studies involving self-reported variables are omnipotent within the fields of public health and, in particular, health promotion. The Authors strongly believe that the current study design was appropriate in the context of our research objectives. In response to Point 3, to ascertain that ‘baseline data analysis would be unworthy’ on this basis would render a significant portion of all published health promotion literature as equally unworthy, including previous research published in the current journal (Varela-Meto et al., 2012; El Ansari et al., 2011).

Finally, we acknowledge the fact that our survey was incentivised with entry to a draw to win an iPad, which may have triggered egoistic participants. However, incentivisation is a commonly cited strategy to increase response rate that has been employed in previous work involving cohorts of higher education students (Kunttu et al., 2012; Hope et al., 2005).  

Point 4: I  will suggest use of dataset from ‘A Healthy CIT’ waiting for 2 or 3 years latter. The authors could think about a smart design to overcome the short response rate and the problem of students egoism.

Response 4: The Authors thank the Reviewer for Point 4. As described in our response to Point 1 above, the current study constitutes a part of a broader programme of research to inform the development of a new campus health promotion initiative. The purpose of the manuscript was to report on baseline findings from a higher education institution in Ireland in order to address the general paucity of empirical Irish evidence to guide health promotion policies and initiatives within HEIs.

The Authors agree with the Reviewer’s recommendation to incorporate future retrospective comparisons with this baseline dataset and this has been  incorporated into the broader research strategy going forward. However, the fundamental purpose of the current study was to investigate the self-reported health metrics and lifestyle behaviours of higher education students, as well as to identify the predictors of positive mental health. The Authors feel that the current cross-sectional study design was appropriate in the context of achieving these research objectives. Similar cross-sectional designs have been reported in a myriad of comparative literature with similar research objectives, including previous work published in the International Journal of Environmental Research and Public Health (Varela-Mato et al., 2012; El Ansari et al., 2011). Longitudinal analyses, although undoubtedly worthwhile, are beyond the scope of the present work but will be a focus of future research activities. 

References

Almutairi, K. M., Alonazi, W. B., Vinluan, J. M., Almigbal, T. H., Batais, M. A., Alodhayani, A. A., ... & Alhoqail, R. I. (2018). Health promoting lifestyle of university students in Saudi Arabia: a cross-sectional assessment. BMC public health18(1), 1093.

Davoren, M. P., Shiely, F., Byrne, M., & Perry, I. J. (2015). Hazardous alcohol consumption among university students in Ireland: a cross-sectional study. BMJ open5(1), e006045.

Dodd, L. J., Al-Nakeeb, Y., Nevill, A., & Forshaw, M. J. (2010). Lifestyle risk factors of students: a cluster analytical approach. Preventive medicine51(1), 73-77.

El Ansari, W., Stock, C., Hu, X., Parke, S., Davies, S., John, J., ... & Mabhala, A. (2011). Feeling healthy? A survey of physical and psychological wellbeing of students from seven universities in the UK. International journal of environmental research and public health8(5), 1308-1323.

Higher Education Authority (n.d.). Search our statistics. Available at: https://hea.ie/statistics-archive/ [accessed 26th October 2019].

Hope, A. Dring, C. & Dring, J. (2005). The health of Irish students: College lifestyle and attitudinal national (CLAN) survey. Department of Health and Children: Dublin, Ireland. 

Kokotailo, P. K., Egan, J., Gangnon, R., Brown, D., Mundt, M., & Fleming, M. (2004). Validity of the alcohol use disorders identification test in college students. Alcoholism: Clinical and Experimental Research28(6), 914-920.

Kwak, N., & Radler, B. (2002). A comparison between mail and web surveys: Response pattern, respondent profile, and data quality. Journal of official statistics18(2), 257.

Kunttu, K., & Pesonen, T. (2013). Student health survey 2012: a national survey among Finnish university students. Helsinki: Finnish Student Health Service.

MacNeela, P.,Dring, C., Ven Lente, E., Place, C., Dring, J  & McCaffrey, J. (2012). Health and wellbeing of NUI Galway undergraduate students: The student lifestyle survey. School of Psychology, Student Services National University of Ireland, Galway, Ireland.

Perusse-Lachance, E., Tremblay, A., & Drapeau, V. (2010). Lifestyle factors and other health measures in a Canadian university community. Applied Physiology, Nutrition, and Metabolism35(4), 498-506.

Robinson, N., Andrews, S., & Yoder, B. E. (2016). Student Lifestyle Choices and Perceptions of Stress Based on Majors. Kinesiology and Allied Health Senior Research Projects 5. Available at: https://digitalcommons.cedarville.edu/cgi/viewcontent.cgi?article=1004&context=kinesiology_and_allied_health_senior_projects

Varela-Mato, V., Cancela, J. M., Ayan, C., Martín, V., & Molina, A. (2012). Lifestyle and health among Spanish university students: differences by gender and academic discipline. International journal of environmental research and public health9(8), 2728-2741.

Reviewer 2 Report

Hello,

I believe this article will be of great usefulness for policy makers in Ireland. I would like to say that I note the hard work you have put in, but I do believe some edits are required:

Line 95: No period at end of sentence

Line 160: Missed space after 'dessert'

Line 249: You indicate in line 247 that participants could opt-out of additional correspondence, but then state in line 249 that three reminder emails were also sent to those who opted-out. Please clarify.

Line 250: Why did you re-open the survey after the semester ended for another week? It seems likely that the variables examined would be quite different between students actively in session vs. on break. 

Line 300: Order here is U, P, Z, R - rest of article seems to be U, Z, P, R. It should be re-structured to be consistent with the rest of the article.

Line 304: The numbers listed don't match what is actually seen in the figure. Please clarify/correct.

Line 316: You say females ate significantly more than fruits and vegetables than males, but the quoted data shows the same Md. value for both (3.0). 

Line 316: Unnecessary period after U value

Line 317: No comma after z value

Line 341: Unnecessary equal sign after U value, no commas between variables

Line 382: Some U values have .0 at the end, this one does not, please be consistent

Line 393: 2x2 is portrayed as both '2 x 2' and '2x2', please be consistent

Line 406: Some 'versus' are displayed as 'vs' and others as 'vs.', please be consistent

Line 427: Space after 'vs.' is required

Line 427: Throughout the article there is inconsistent spacing around '=', please be consistent

Line 490-493: You state that previous literature suggests females in the overweight category will perceive themselves to be normal weight or fail to recognize the extent of their overweight. However, your results differ from this previous literature and no explanation is provided. Please expand.

Line 534: You say trends, but don't discuss what those trends are, so it is unclear. Did males previously have similar drug use patterns as females, but now are increasing? I think it would be more accurate to simply say that your data is consistent with other literature. 

Line 557: Circadian should not be capitalized, and there are more than one type of circadian rhythm that can stabilised (i.e., make it plural).

It has been a pleasure reading this work. I appreciate your time and efforts!

Author Response

Response to Reviewer 2 Comments

Hello,

I believe this article will be of great usefulness for policy makers in Ireland. I would like to say that I note the hard work you have put in, but I do believe some edits are required:

Point 1: Line 95: No period at end of sentence

Response 1: The Authors thank the Reviewer for Point 1. The missing period has been inserted.

Point 2: Line 160: Missed space after 'dessert

Response 2: The Authors thank the Reviewer for Point 2. The missed space has been inserted.

Point 3: Line 249: You indicate in line 247 that participants could opt-out of additional correspondence, but then state in line 249 that three reminder emails were also sent to those who opted-out. Please clarify

Response 3: The Authors thank the Reviewer for Point 3 and understand that this should have been presented more clearly. Study participants who chose to opt out of correspondence did not receive subsequent communications regarding the study (including reminder emails) from that time point forward. In order to reflect this, the words ‘or opted out of correspondence’ have been removed (see Line 254 in revised manuscript). In addition, Lines 252-253 have been amended to state ‘Participants could also opt-out of any further correspondence, if desired, and those who opted out received no further contact.’

Point 4: Line 250: Why did you re-open the survey after the semester ended for another week? It seems likely that the variables examined would be quite different between students actively in session vs. on break.

Response 4: The Authors thank the Reviewer for Point 4 and acknowledge that this is a pertinent point of discussion given the self-report nature of the study. The Authors believe that the decision to re-open the survey for a 7 day period did not compromise the validity of data collection for the following reasons:

  1. Firstly, the higher education setting in question (Cork Institute of Technology) delivers its academic programmes via a semesterised model. The academic year is comprised of two equally weighted semesters (Oct-Dec and Feb-May). In addition to designated examination periods at the end of each semester, continuous assessment is an inherent feature of this organisational structure. Assessment tends to take place continuously throughout the Semester within a modular system. In addition, many modules are assessed on an entirely continuous basis, with no formal end of semester written examination. Therefore, the in-semester demands placed on students in terms of workload and assessment are not confined solely to examination periods and the academic workload, commitment and burden is dispersed relatively evenly throughout the year. In this regard, the Authors believe that any alleged workload-related differences in reported health and wellbeing variables between respondents who participated prior to the examination periods (which consisted of a non-contact ‘Reading Week’) versus the limited 7 day period directly following the written examination period would be minimal.
  2. Even if alleged short-term (the time point in question was a limited 7 day window post exam period) differences in self-reported health metrics and lifestyle variables existed, the recall time frame of pertinent validated scales and questionnaire items utilised in this study would attenuate this. For example, the Mental Health Index-5 (MHI-5) and the Energy and Vitality Index (Ware et al., 1993), which were measures of negative and positive mental health respectively, request a recall of experiences over the previous 4 weeks. Similarly, the Alcohol Use Disorders Identification Test (AUDIT) (Babor et al., 2001) enquires predominantly about habitual alcohol behaviours, with a number of items referring to behaviours over the ‘past year’.
  • From an ethical perspective it was deemed appropriate to cease email reminder/communications and prevent students accessing the survey during the formal examination period. This was not because the Authors’ believed that there would be huge variability in responses, but rather that it prevented a student using designated study time to complete the survey. In addition, it prevented the (albeit unlikely) scenario of a students’ exam performance being impaired because they were acutely affected by any of the issues raised in the survey if they decided to complete it in the immediate hours prior to an exam.
  1. Finally, the Authors received anecdotal feedback from prospective student participants that they would like to contribute to the survey following the cessation of their examinations, given that findings would be used to inform internal health promotion initiatives and policy. Therefore, it was decided to provide any such student with a final opportunity to participate.

Although the Authors feel that this methodological decision was justified and would not have compromised validity, a statement acknowledging the potential limitations has been added section 4.2 (Limitations); Lines 679-687. The Authors once again thank the Reviewer for bringing this Point to our attention.

Point 5: Line 300: Order here is U, P, Z, R - rest of article seems to be U, Z, P, R. It should be re-structured to be consistent with the rest of the article

Response 5: The Authors thank the Reviewer for Point 5. The order has therefore been amended as suggested to U, Z, P, R in order to be consistent with the remainder of the article.

Point 6: Line 304: The numbers listed don't match what is actually seen in the figure. Please clarify/correct

Response 6: The Authors thank the Reviewer for Point 6. As described in Lines 260-261, all categorical data is presented relative to the number of valid responses received to each applicable item. The values presented in Line 304-305 describe the relative distribution of total valid responses received to the item regarding self-perceived weight category.

However, Figures 1 and 2 directly compare the self- perceived versus calculated BMI categories. These figures include only those participants who provided all necessary data (i.e. those who answered both the self-perceived BMI item and also provided a valid height and body mass figure to enable BMI calculation). This explains why the numbers referred to by the Reviewer do not coincide; they are effectively independent of each other.

Point 7: Line 316: You say females ate significantly more fruits and vegetables than males, but the quoted data shows the same Md. value for both (3.0).

Response 7: The Authors wish to outline that the significance of any Mann Whitney U test between two groups does not depend specifically on the median values (Hart, 2001).  Statistically, the distribution or ‘spread’ of data within a group also exerts an influence and it is possible for the null hypothesis to be rejected even if median values are equal (Hart, 2001). The rejection of the null hypothesis was based on ‘mean rank’ values calculated for each group. The median value was reported to provide an indication of the central value within each group which the Authors felt would constitute a more meaningful metric in terms of conceptualising daily fruit and vegetable servings. However, the Authors understand the confusion this may have caused due to the identical median values and therefore have amended Lines 324 and 325 to include mean rank values instead. The ‘alternative hypothesis’ was that, based upon ranked values, there was more than a 50-50 probability that a reported serving value in the female group would be higher than a value in the male group (Nachar, 2008)

Point 8: Line 316: Unnecessary period after U value

Response 8: The Authors thank the Reviewer for Point 8. The period has been removed (see Line 325 in amended manuscript).

Point 9: Line 316: No comma after z value

Response 9: The Authors thank the Reviewer for Point 9. The comma has been inserted (see Line 325 in amended manuscript).

Point 10: Line 341: Unnecessary equal sign after U value, no commas between variables

Response 10: The authors thank the Reviewer for this Point. The unnecessary equal sign has been removed and commas have been inserted where necessary (see Line 361 in amended manuscript)

Point 11: Line 341: Some U values have .0 at the end, this one does not, please be consistent

Response 11: The Authors thank the Reviewer for this Point. All U values have been removed and/or amended to ensure they are presented consistently with a single decimal (e.g. ‘.0’) at the end.

Point 12 Line 393: 2x2 is portrayed as both '2 x 2' and '2x2', please be consistent

Response 12: The Authors thank the Reviewer for highlighting this. The erroneous formatting has been corrected to ‘2x2’ (see Line 427 on the amended manuscript).

Point 13: Line 406: Some 'versus' are displayed as 'vs' and others as 'vs.', please be consistent

Response 13: The Authors thank the Reviewer for this Point. The terms in question were reviewed to ensure consistency throughout the document. It was decided that ‘vs.’ would be used consistently and any ‘vs’ values were amended to ‘vs.’ throughout the document. In addition ‘versus’ has been amended to ‘vs.’ in the headings of both Figure 1 and Figure 2. ‘Versus’ has also been changed to ‘vs.’ in the column heading of Table 4.

Point 14: Line 427: Space after 'vs.' is required

Response 14: The Authors thank the Reviewer for bringing this to our attention. A space has been inserted (this is now positioned in Line 468 in the amended manuscript) 

Point 15: Line 427: Throughout the article there is inconsistent spacing around '=', please be consistent

Response 15: The Authors acknowledge the inconsistent spacing and have reviewed and amended the entire document. There are now no spaces immediately after the ‘=’ symbol.

Point 16: Line 490-493: You state that previous literature suggests females in the overweight category will perceive themselves to be normal weight or fail to recognize the extent of their overweight. However, your results differ from this previous literature and no explanation is provided. Please expand.

Response 16: The Authors thank the Reviewer for this Point and understand the need for expansion and clarification. In Lines 546-547 the following statement was included in the original manuscript: “Strong evidence has accumulated to suggest that a significant proportion of those who are overweight or obese either perceive themselves to be of normal weight or fail to recognise the extent of their overweight/obesity”. The purpose of this sentence, which appears to have caused confusion, was not to make any statement regarding females in particular, but to present a key overall finding reported in a comprehensive literature review (Robinson et al., 2017). This work referred clearly to the significant body of evidence to support the fact that a substantial proportion of individuals (male and female) who are overweight or obese tend to underestimate their weight category. ‘Visual normalisation’ was proposed as a contributing mechanism whereby, as overweight/obesity prevalence continues to rise, individuals internalise an inaccurate (i.e. larger) perception of what constitutes a ‘normal’ BMI weight category. However, an additional point also noted in this work was that, although underestimation of overweight/obesity appears to be common among both sexes, females are less likely to underestimate. This may be due to females’ increased exposure to slender/lean body types in popular media, attenuating the effects of visual normalisation to a certain extent.

With regard to the findings of the current study, as outlined in Lines 549-550, both sexes underestimated the prevalence of obesity (see Figures 1 and 2). This was consistent with the generalised position stand of Robinson et al. (2017) referred to above. However, with regard to the ‘overweight’ category, although males still tended to underestimate (Figure 2; 22.4% perceived vs. 31.4% calculated prevalence) females, in fact, overestimated ‘overweight’ (30.7% perceived prevalence vs 23.0% calculated). In the case of females only, this contrasts with the overall consensus of Robinson et al., but does substantiate their suggestion that females may be less likely to underestimate overweight due to increased visual exposure to leaner body types in popular media. As a finding, it clearly requires further exploration as already outlined in Lines 561-565.  

In order to reflect  these interesting findings more clearly, the original wording of Lines 546-565 has been amended to expand on the Reviewer’s feedback and the Authors hope this clarifies matters sufficiently.

Point 17: Line 534: You say trends, but don't discuss what those trends are, so it is unclear.

Did males previously have similar drug use patterns as females, but now are increasing? I think it would be more accurate to simply say that your data is consistent with other literature.

Response 17: The Authors appreciate that the use of the word ‘trends’ created ambiguity with regard to this statement. The data is consistent with previous research involving undergraduate university students in Ireland (Hope et al., 2005) whereby a significantly greater proportion of males reported illicit drug use in comparison to females. The reference to ‘trends’ has therefore been removed and replaced with an alternative statement that confirms the current data is consistent with previous literature (Line 597)

Point 18: Line 557: Circadian should not be capitalized, and there are more than one type of circadian rhythm that can stabilised (i.e., make it plural).

Response 18: The Authors thank the Reviewer for this Point. The respective capital letter has been removed from ‘circadian’ (Line 624 in amended manuscript) and the word ‘rhythm’ has been amended to ‘rhythms’.

It has been a pleasure reading this work. I appreciate your time and efforts!

Once again, the Authors wish to sincerely thank Reviewer 2 for such a comprehensive and constructive review of this manuscript. We hope that we have addressed all Points as outlined above.

References

Babor, T. F., de la Fuente, J. R., Saunders, J., & Grant, M. (2001). The Alcohol Use Disorders Identification Test: Guidelines for use in primary care.

Hart A. (2001). Mann-Whitney test is not just a test of medians: differences in spread can be important. BMJ (Clinical research ed.)323(7309), 391–393. doi:10.1136/bmj.323.7309.391

Hope, A. Dring, C. & Dring, J. (2005). The health of Irish students: College lifestyle and attitudinal national (CLAN) survey. Department of Health and Children: Dublin, Ireland.

Nachar, N. (2008). The Mann-Whitney U: A test for assessing whether two independent samples come from the same distribution. Tutorials in quantitative Methods for Psychology4(1), 13-20.

Robinson, E. (2017). Overweight but unseen: a review of the underestimation of weight status and a visual normalization theory. Obesity Reviews18(10), 1200-1209.

Ware, J., Snoww, K.K., Kosinksi, M.A., Gandek, B.G. (1993). SF-36 Health Survey: Manual and Interpretation Guide. Lincoln, RI: Quality Metric, Inc,

Reviewer 3 Report

75. Did students state that they had a lack of concern regarding STIs or is this presumed do to the increase of "hooking up"?  

478.  Elaborate on the discrepancy in perceived vs. actual BMI of males.  Was the difference greater in low or high BMI?

Author Response

Response to Reviewer 3 Comments

The Authors would like to thank Reviewer 3 for the review points below and for the positive evaluation of the submitted manuscript. Responses to each of the specific points raised are outlined in further detail below.

Point 1: 75. Did students state that they had a lack of concern regarding STIs or is this presumed do to the increase of "hooking up"?

Response 1: The Authors thank the Reviewer for Point 1. In clarification, Line 75 was in reference to a reported finding from a qualitative study (Downing-Matibag & Geisinger, 2009) that explored various elements of sexual risk taking during ‘hook up’ encounters (an encounter could involve intercourse and/or other sexual acts). Data was collected from students at a Midwestern US university via semi-structured interviews. A section of the interview was specifically focussed on the assessment of students’ perceptions of their risk taking behaviours in the context of sexually transmitted infections (STIs). Students were asked whether they had been previously tested for, or whether they were concerned about contracting an STI. Following analysis and thematic coding of data, it emerged that students were generally unaware of their susceptibility to STIs. Merely half of students felt concerned about contracting an STI following a sexual ‘hook-up’ that involved intercourse. Furthermore, the majority of students were not concerned about contracting an STI as a result of a sexual ‘hook-up’ that did not result in full intercourse, but involved some other sexual act.

Point 2: 478.  Elaborate on the discrepancy in perceived vs. actual BMI of males.  Was the difference greater in low or high BMI?

Response 2: The Authors thank the Reviewer for this Point and welcome the opportunity to elaborate on these interesting findings. The Authors refer the Reviewer specifically to Figure 2 for a visual representation of this data. In the ‘underweight’ BMI category (i.e. corresponding to the lowest BMI values of less than 18.5 kg/m2) the difference between perceived versus calculated prevalence was 4.5% (6.9% perceived versus 2.4% calculated). Within the normal range of BMI values (18.5-24.9 kg/m2), the difference between perceived (68.9%) versus actual prevalence (55.5%) was 13.4%. In the ‘overweight’ (BMI 25.0-29.9 kg/m2) and ‘obese’ (BMI greater than 30.0 kg/m2) categories, the direction of the discrepancies reversed (i.e. a greater proportion of males were calculated to reside within each of these categories versus the respective perceived proportions). In terms of the absolute differences; perceived (22.4%) versus calculated (31.4%) overweight prevalence differed by 9%, and perceived versus calculated obesity prevalence (1.8% vs. 10.7%) differed by 8.9%.

In order to compare the true magnitude of the differences between perceived and actual BMI of males across each of the BMI categories (as requested by the Reviewer), the numerical value of the percentage difference (x) was expressed relative to the calculated prevalence (y) within each category (x/y*100). Using this method, it appears that ‘underweight’ was overestimated by 188% (4.5/2.4*100), ‘normal weight’ was overestimated by 24% (13.4/55.5*100), ‘overweight’ was underestimated by 29% (9/31.4*100) and ‘obesity’ underestimated by 83% (1.8/10.7*100). Therefore, it appears that, for males, the magnitude of under/over estimation of perceived vs. calculated BMI category is greater as one moves towards a very low (underweight) or very high (obese) BMI and is less exaggerated for the normal weight and overweight categories. This point has been expanded and included in the revised manuscript (Lines 534-538)

References

Downing-Matibag, T. M., & Geisinger, B. (2009). Hooking up and sexual risk taking among college students: A health belief model perspective. Qualitative Health Research19(9), 1196-1209.

Round 2

Reviewer 1 Report

Since the authors reply my previous comment in detail, however, they still did not answer how the 21.2% response rate (actually, removing missing and spoiled data, only 20.1% total response rate) would be representative all registered students at Cork Institute of Technology(n=11,261) in Semester Two (Spring) of a standard academic year in 2016. At least, the authors can check that it skewed in some obtained variables, or not?

On the other hand, because the answers of college students may skew to self-interest (egoism), I think the baseline (2016) data should be followed up in 2 or 3 years later, and it should be more valuable to ‘Healthy CIT’. It means to add data in 2018 or later should be more valuable.

Author Response

Response to Reviewer 1 Comments

Point 1: Since the authors reply my previous comment in detail, however, they still did not answer how the 21.2% response rate (actually, removing missing and spoiled data, only 20.1% total response rate) would be representative all registered students at Cork Institute of Technology(n=11,261) in Semester Two (Spring) of a standard academic year in 2016. At least, the authors can check that it skewed in some obtained variables, or not?

Response 1: The Authors thank the Reviewer for Point 1. In the first instance, we would like to refer the Reviewer to Table 1 (beginning Line 293) and particularly focus on the breakdown of student participants by ‘Faculty’. As a HEI, Cork Institute of Technology is comprised of a primary central campus that houses the two largest faculties (Business & Humanities and Engineering & Science), in addition to a number of smaller satellite campuses (Crawford College of Art and Design, Cork School of Music, and the National Maritime College of Ireland) at other geographical locations across Cork city and county. As outlined in Table 1, our dataset comprised of students from all campuses of the Institute. Furthermore, in keeping with the proportional representation of the Institute, the most significant representation (42.9% and 42.0%) was from the two largest central faculties (Business & Humanities and Engineering & Science).

In terms of whether individualised variables were ‘skewed’, the Authors are assuming that the Reviewer is referring to potential bias introduced by our convenience sampling method. The Author’s agree with the Reviewer that it is possible that data was not representative of the student population as a whole. However, as previously stated, this is a generalised methodological limitation and not confined to our study. This limitation has been overtly acknowledged since the very first iteration of the manuscript in Lines 703-704 where it states ‘although all students were eligible to participate, a convenience sample was employed, which potentially introduced a selection bias’.

Therefore, as with any self-report, survey-based study incorporating a convenience sample, it may indeed be the case that the data may not be generalizable to the Institute’s population as a whole. This has been extensively acknowledged in previous work (El Ansari et al., 2011; Dodd et al., 2010; Keller et al., 2008). However, in order to reflect this more clearly within the current manuscript, the Authors have amended Lines 704-705 to state ‘Therefore, findings may not be representative of all students of the Institute’. Within the scope of the current paper, it is not feasible to calculate the extent to which each of the variables in our comprehensive measurement instrument may or may not have been skewed in any particular manner. Given the extensive body of peer-reviewed research published employing a similar methodological design, as well as the fact that this known limitation has been repeatedly acknowledged in the manuscript, we hope that this will satisfy the Reviewer’s legitimate concerns in this regard. 

In addition, and as outlined previously, our response rate needs to be interpreted in the context of similar work in the field. A novel aspect of our research was that, because of our selected survey platform, all registered students (n=11,261) were given an equal opportunity to participate. The Authors understand that, although our sample size/response rate was discussed extensively in our previous response to Reviewer 1, this information should have been incorporated directly into the manuscript itself. Therefore, a discussion of how our 21.2% response rate relates to previous work has now been included (Lines 511-534). The Authors once again thank the Reviewer for bringing this to our attention and for suggesting this addition to enhance the quality of our manuscript.

Point 2: On the other hand, because the answers of college students may skew to self-interest (egoism), I think the baseline (2016) data should be followed up in 2 or 3 years later, and it should be more valuable to ‘Healthy CIT’. It means to add data in 2018 or later should be more valuable

Response 2: The Authors thank the Reviewer for once again raising this point. As outlined previously, reliance on self-reported data will always incur limitations. However, in the context of the field of health promotion, self-reported data forms the basis of the majority of published work. The Authors acknowledge that we may not have sufficiently addressed the Reviewer’s concerns regarding egoism. Therefore, we have amended the wording of Lines 705-706 of the manuscript to state ‘As there was an inevitable reliance on self-report data, under/over-reporting, egoism and socially-desirable responses cannot be ruled out’.

Regarding conducting a follow-up study, the authors wish to reiterate that the current study constitutes a part of a broader programme of research and a follow-up study is currently being planned. Data will be collected in the manner recommended by the Reviewer in order to facilitate comparisons with baseline data. Although beyond the scope of the current paper, this planned work has been described in the revised manuscript, Lines 697-700. The Authors once again thank the Reviewer for the recommendations in this regard.

References

Dodd, L. J., Al-Nakeeb, Y., Nevill, A., & Forshaw, M. J. (2010). Lifestyle risk factors of students: a cluster analytical approach. Preventive medicine, 51(1), 73-77. 

El Ansari, W., Stock, C., Hu, X., Parke, S., Davies, S., John, J., ... & Mabhala, A. (2011). Feeling healthy? A survey of physical and psychological wellbeing of students from seven universities in the UK. International journal of environmental research and public health, 8(5), 1308-1323. 

Keller, S., Maddock, J. E., Hannöver, W., Thyrian, J. R., & Basler, H. D. (2008). Multiple health risk behaviors in German first year university students. Preventive medicine46(3), 189-195.